# Biocompatibility Assessment of Polycaprolactone/Polylactic Acid/Zinc Oxide Nanoparticle Composites Under In Vivo Conditions for Biomedical Applications

**DOI:** 10.3390/pharmaceutics15092196

**Published:** 2023-08-25

**Authors:** Jorge Iván Castro, Daniela G. Araujo-Rodríguez, Carlos Humberto Valencia-Llano, Diego López Tenorio, Marcela Saavedra, Paula A. Zapata, Carlos David Grande-Tovar

**Affiliations:** 1Laboratorio SIMERQO, Departamento de Química, Universidad del Valle, Calle 13 No. 100-00, Cali 76001, Colombia; jorge.castro@correounivalle.edu.co; 2Grupo de Investigación de Fotoquímica y Fotobiología, Universidad del Atlántico, Carrera 30 Número 8-49, Puerto Colombia 081008, Colombia; dgaraujo@mail.uniatlantico.edu.co; 3Grupo Biomateriales Dentales, Escuela de Odontología, Universidad del Valle, Calle 4B # 36-00, Cali 76001, Colombia; carlos.humberto.valencia@correounivalle.edu.co (C.H.V.-L.); diego.lopez.tenorio@correounivalle.edu.co (D.L.T.); 4Grupo de Polímeros, Facultad de Química y Biología, Universidad de Santiago de Chile, Santiago 9170020, Chile; alecram.saavedra@gmail.com (M.S.); paula.zapata@usach.cl (P.A.Z.)

**Keywords:** biodegradation, biomaterial, biomedical applications, ZnO-NPs, nanocomposites

## Abstract

The increasing demand for non-invasive biocompatible materials in biomedical applications, driven by accidents and diseases like cancer, has led to the development of sustainable biomaterials. Here, we report the synthesis of four block formulations using polycaprolactone (PCL), polylactic acid (PLA), and zinc oxide nanoparticles (ZnO-NPs) for subdermal tissue regeneration. Characterization by Fourier transform infrared spectroscopy (FT-IR) and X-ray diffraction (XRD) confirmed the composition of the composites. Additionally, the interaction of ZnO-NPs mainly occurred with the C=O groups of PCL occurring at 1724 cm^−1^, which disappears for F4, as evidenced in the FT-IR analysis. Likewise, this interaction evidenced the decrease in the crystallinity of the composites as they act as crosslinking points between the polymer backbones, inducing gaps between them and weakening the strength of the intermolecular bonds. Thermogravimetric (TGA) and differential scanning calorimetry (DSC) analyses confirmed that the ZnO-NPs bind to the carbonyl groups of the polymer, acting as weak points in the polymer backbone from where the different fragmentations occur. Scanning electron microscopy (SEM) showed that the increase in ZnO-NPs facilitated a more compact surface due to the excellent dispersion and homogeneous accumulation between the polymeric chains, facilitating this morphology. The in vivo studies using the nanocomposites demonstrated the degradation/resorption of the blocks in a ZnO-NP-dependant mode. After degradation, collagen fibers (Type I), blood vessels, and inflammatory cells continue the resorption of the implanted material. The results reported here demonstrate the relevance and potential impact of the ZnO-NP-based scaffolds in soft tissue regeneration.

## 1. Introduction

Disease, trauma, and injury can damage the human body, impacting patients’ quality of life and necessitating costly tissue repair, replacement, and regeneration treatments. Traditional approaches involve autografts (from the same patient) or allografts (from another individual), but both methods have significant drawbacks. Autografts involve expensive, painful, and challenging tissue harvesting, leading to donor site morbidity. Allografts can be rejected by the patient’s immune system and carry the risk of disease spreading from the donor [1,2].

Currently, 3D porous scaffolds are used with the ability to provide the right environment for tissue regeneration [3]. Generally, scaffolds are developed from organic and inorganic materials with cells or growth factors to combine the particular benefits of each component, enhancing cellular response (adhesion, proliferation) as well as the mechanical, physical, thermal, biocompatible, and biodegradable properties required in many tissues and biomedical applications [4]. In this sense, polymeric matrices (natural or synthetic) reinforce the promotion of tissue regeneration [3].

Polylactic acid (PLA) is derived from lactic acid obtained by fermentation of various biological sources of lactic acid bacteria. This polymer substitutes the olefinic plastics. The easy accessibility from natural resources, biocompatibility, and biodegradability introduce broad interest in biomedical applications [5]. Thus, PLA is used in tissue engineering to restore or improve tissue functions by combining bioactive molecules, cells, and scaffolds. PLA possesses biocompatible properties that facilitate scaffolds with optimal architectures for the regeneration of complex tissues [6]. However, adding soft and ductile polymers such as polycaprolactone, polyurethane, or chitosan improves the poor mechanical properties of PLA [7].

Polycaprolactone (PCL) is a thermoplastic, biodegradable, and biocompatible synthetic polymer derived from ε-caprolactone, with an amorphous phase at ambient temperature and melting temperature around 55–70 °C [8]. PCL is synthesized from a ring-opening mechanism using a catalyst [9]. However, PCL/PLA blends present mechanical problems at elevated temperatures due to the slow crystallization rate of PLA, which is related to the enantiomeric mixture of the major component (the L isomer) and the glass transition temperature, which usually ranges between 55 and 65 °C [10,11]. Therefore, research to counteract this drawback of mixtures has focused on the addition of ceramic nano-fillers, i.e., inorganic nanoparticles.

Zinc oxide nanoparticles (ZnO-NPs) are suitable semiconductors with high chemical stability and low cost, whose safe application is supported by the Food and Drug Administration (FDA) [12]. ZnO-NPs are synthesized by precipitation [13], sol-gel method [14], spray pyrolysis [15], thermal decomposition [16], and forced solvolysis [17]. Each method has both advantages and disadvantages. For example, additional procedures such as calcination to prepare ZnO-NPs from the precipitation method [18], disposition of raw materials from the natural extracts method, nanodomain overlay by the spray pyrolysis method [19], and production of zinc acetate by the forced solvolysis method [20]. However, the latter has allowed morphological control depending on the type of alcohol and the number of carbons present. For example, methanol resulted in nanospheres, while polyhedral morphology was obtained when using 1-butanol and becoming rod-shaped for 1-hexanol [17].

In the literature, several investigations analyze the influence of ZnO-NPs in the medicinal field due to their excellent antimicrobial activity against a wide range of microorganisms, such as bacteria and fungi [21,22]. The mechanism of the antimicrobial action of ZnO-NPs is not described. However, there are two separate pathways [17,23]. In the presence of light, ZnO-NPs produce reactive oxygen species (ROS) responsible for the oxidative stress produced in the microorganisms across the cell membrane. On the other hand, under dark conditions, ZnO-NPs mechanically pierce the cell membrane and interfere with the essential constituents of the microbial cell. They have also been studied in the development of Zn-dependent matrix metalloproteinases due to their ability to degrade almost all components of the extracellular matrix responsible for cellular repair and tissue remodeling [24]. Due to their potent antimicrobial activity, ZnO-NPs have been studied in topical ointments [25], controlled-release drugs [26], tissue dressings [27], and as components in antimicrobial packaging [28]. Some materials such as glass, paper, chitosan, polyethylene, polypropylene (PP), polyurethane (PU), and low-density polyethylene have been coated with ZnO-NPs to obtain biodegradable materials with antimicrobial activity [29,30].

Several studies have been conducted for systems containing the tricomponent mixture PCL/PLA/ZnO-NPs, where the effect of using ZnO-NPs for therapeutic studies has been scarcely explored. The use of biodegradable polymeric matrices for a controlled release of ZnO-NPs for specific treatments has been explored in several investigations [31,32,33]. Our research group has implemented the formation of membranes using components with biodegradable and biocompatible properties as polymeric matrices. These studies found that adding tea tree essential oil (TTEO) accelerated the degradative processes without necrosis due to the absence of pus and decreased inflammatory infiltrate after 90 days of implantation [34,35].

However, no PCL/PLA/ZnO-NP scaffolds have been studied in animal models as a first approximation to current medical devices. In this sense, in this work, we propose the formulation of four PCL/PLA/ZnO-NP blocks to be evaluated subdermally after 60 days of implantation in Wistar rats to observe the material’s biocompatibility. The biocompatibility results showed the presence of connective tissue constituted mainly by type I collagen for all the formulations; nevertheless, for F4, a smaller arrangement of these type I fibers was observed, which is related to an improvement in the healing process probably due to the increase in the concentration of ZnO-NPs as well as to the decrease in PCL.

## 2. Materials and Methods

### 2.1. Materials

Perstorp Company supplied PCL with 6800 layers and a molecular weight of 8000 Da from Warrington, UK. The PLA contains a molecular weight of 20,000 Da with 2% D-isomer supplied by Nature Work from Minneapolis, MN, USA. ZnO-NP synthesis used 80% ZnCl_2_, 99% sodium hydroxide, and 99.8% 2-propanol provided by Merck from Burlington, MA, USA).

### 2.2. Preparation of ZnO-NPs

The ZnO-NPs were synthesized following previously reported methodologies [36]. First, a 0.2 M aqueous solution of ZnCl_2_ and a 5 M NaOH solution were prepared using Milli Q water. The NaOH solution was slowly added dropwise while maintaining a temperature of 90 °C under vigorous stirring for 10 min, resulting in a white suspension. The excess NaCl particles were removed by washing the mixture with ample Milli Q water. The ZnO-NP dispersion was carried out using an ultrasonic bath was mixed with 2-propanol at room temperature for eight minutes (r.t., 10 min). The resulting product was centrifuged at 5000 rpm for 20 min and washed using 2-propanol. Finally, the obtained product was calcined at 250 °C for 5 h.

### 2.3. Preparation of PCL/PLA/ZnO-NP Blocks

The PCL/PLA/ZnO-NP blocks were prepared following previous methodologies [34]. To prepare the blocks, different amounts of PCL, PLA, and ZnO-NPs were added according to Table 1, keeping the presence of 4% by weight of the total formulation constant. Firstly, ZnO-NPs were sonicated at 10 mg/mL concentrations for two hours. Next, each component was dissolved in chloroform and mixed in a glass beaker. The resulting mixture was subjected to an additional two hours of ultrasonic treatment in the bath to eliminate bubbles from the solution. Then, the mixture was drop-casted onto glass molds to form the PCL/PLA/ZnO-NP blocks and evaporated at room temperature for 24 h. After this process, the final PCL/PLA/ZnO-NP blocks were obtained.

### 2.4. Characterization of the ZnO-NPs

A JEOL ARM 200 F transmission electron microscope (TEM) from Tokyo, Japan, operated with a Schottky field emitter with a voltage of 20 kV, was applied to acquire the images of the ZnO-NPs. These were mixed in ethanol and placed on a carbon-coated copper grid with 400 mesh for sample preparation. The solvent was then sonicated and evaporated. The average size of the ZnO-NPs was determined by analyzing 100 nanoparticles using Image J 1.49q software. Attenuated Total Reflectance-Fourier Transform Infrared Spectroscopy (ATR-FTIR) with a diamond tip was used to study the functional groups present in ZnO-NPs using an FT-IR-8400 instrument from Shimadzu, Kyoto, Japan, in the range 4000–500 cm^−1^ with a spectral resolution of 4 cm^−1^ at 32 scans for each spectrum. X-ray diffraction (XRD) measurements were carried out using a PANalytical X0Pert PRO diffractometer instrument from Malvern Panalytical, Royston, UK, with a 1.54 Cu source operated in the secondary electron mode at 45 kV where scanning was performed from 5 to 80°. The scan rate was 2 degrees/min, the scan speed was 2.63 s, and the step size was 0.01°. The whole characterization of the ZnO-NPs was described in previous work [34]. The crystallite size was determined using the Scherrer formula.

(1)
$$\tau =\frac{K\lambda }{\beta \cos\left(\theta \right)}$$


The average crystallite size of the ZnO-NPs was determined using the diffraction peaks (2θ) 32, 34, and 36 attributed to the (100), (002), and (101) planes and Scherrer’s formula. K is the Scherrer constant, and the crystalline shape factor is 0.89. The X-ray source wavelength (λ) was 1.5405 Å. The full width at half maximum of the diffraction peak (β) and the Bragg angle (θ) of the intense peak were also considered. The calculated crystallite size was 30.13 nm.

### 2.5. Characterization of PCL/PLA/ZnO-NP Blocks

#### 2.5.1. Fourier Transform Infrared Spectroscopy and X-ray Diffraction

The FTIR spectra and the diffractogram were obtained under the same conditions described for the characterization of the ZnO-NPs.

#### 2.5.2. Thermal Analysis of the PCL/PLA/ZnO-NP Blocks

A NETZSCH TG 209 F1 Libra instrument from Mettler Toledo, Schwerzenbach, Switzerland was used to determine the thermal properties of the PCL/PLA/ZnO-NP blocks. The samples were heated from 25 to 900 °C in an Al_2_O_3_ crucible at 10 °C/min in a nitrogen atmosphere. The thermodynamic parameters were analyzed by differential scanning calorimetry (DSC) on a DSC1/500 instrument (Mettler Toledo, Schwerzenbach, Switzerland) in a nitrogen atmosphere. The thermal analysis is programmed taking into account a cross-section of 9 mg of the blocks heating from 25 to 250 °C at a rate of 10 °C/min, allowing the glass transition temperature (T_g_) calculated with the midpoint of the transition, the melting temperature (T_m_) as the endothermic peak, and the crystallization temperature (T_cc_) as the exothermic peak to be obtained. The measurements were obtained from the second heating to ensure the polymer’s thermal memory was eliminated. TA Instruments Universal Analysis Software 2000 version 4.5A was used to analyze the thermal properties.

The polymer’s percentage crystallinity was calculated by Equation (2) [37].

(2)
$$X_{c}=\frac{\left(\Delta H_{m}-\Delta H_{CC}\right)}{\Delta H_{m}^{{}^{\circ}}\left(1-x\right)}$$

where 
$\Delta H_{m}^{{}^{\circ}}$
 is the enthalpy of fusion of pure PLA at 100% crystallinity [38]. Meanwhile, 
$\Delta H_{m}$
 and 
$\Delta H_{CC}$
 stand for the enthalpy of fusion and enthalpy of crystallization in J/g, respectively. The weight percent of PLA in the nanocomposites is denoted by the term (1 − *x*).

#### 2.5.3. Microstructure Studies

Morphological analysis was performed by scanning electron microscopy using a Hitachi^TM^ 3000 scanning electron microscope in the secondary electron mode at 20 kV. To enhance sample conductivity, a gold coating was applied before imaging.

### 2.6. In Vivo Biocompatibility Studies of the PCL/PLA/ZnO-NP Blocks

#### 2.6.1. Surgical Preparation of Biomodels

The study aimed to assess the in vivo biocompatibility of the four formulations using a subcutaneous implantation design. This design allowed multiple samples to be implanted simultaneously in biomodels, and their behavior was studied over time [39]. The experiment followed the guidelines of UNE: 10993-6 (Biological evaluation of medical devices—Part 6: Tests for local effects after implantation. ISO 10993-6: 1994). Three biomodels (Wistar rats, five-month-old males, with a 380 g weight) were selected from the Intermediate Laboratory of Preclinical Research and Biotherium (LABBIO) of the Universidad del Valle. After incising the biomodels, the different blocks were implanted to test the preliminary biocompatibility. Three pockets, each 1 cm wide and 10 cm deep, were created with hemostatic forceps, with a centimeter of separation between them [40].

Sedation of the biomodels consisted of Ketamine (70 mg/kg, Blaskov Laboratory, Bogotá, Colombia) and Xylazine (30 mg/kg, ERMA Laboratories, Celta, Colombia). The implantation site was isolated and disinfected. Anesthesia was then applied locally using Lidocaine with epinephrine (2%). Four 5-mm-long incisions were made along the dorsal midline, one centimeter apart. Subcutaneous pocket-type preparations were created on the right side, 2 cm deep, using hemostatic forceps.

After 60 days of implantation, the biomodels were euthanized using intraperitoneal injection of sodium pentobarbital/sodium diphenylhydantoin. Macroscopic observation of the hair-bearing implantation areas was performed, and trichotomy was carried out to recover the samples. The samples were stored in plastic bottles with buffered formalin, followed by several washings with phosphate-buffered saline, dehydration (ascending alcohol concentrations), xylol diaphanization, and infiltration with paraffin using the Auto-technicon Tissue Processor™ of Leica Microsystem from Mannheim, Germany. The cuts were made using the Thermo Scientific™ Histoplast Paraffin™ kit provided by Fisher Scientific from Waltham, MA, USA.

#### 2.6.2. Histological Analysis

After obtaining the paraffin blocks with the tissue samples, they were cut to a thickness of 5 µm using the Leica microtome instrument. After 48 h on a slide, histological analysis was performed using hematoxylin-eosin (HE) and Masson’s trichotomy (MT) stains. Histological images were captured using a Leica DFC 295 camera and a Leica DM750 optical microscope. Image interpretation was performed using Lecia Microsystems 4.12.0 software provided by Leica Microsystem (Mannheim, Germany).

Ethical approval and supervision were performed by the Biomedical Experimentation Animal Ethics Committee (CEAS) of the Universidad del Valle. The study followed the “Animal Research: Reporting of In Vivo Experiments” (ARRIVE) guidelines [41]. No intraoperative or postoperative complications or deaths of biomodels occurred during the research. Control material was not used to avoid additional discomfort to the animals, as the aim was to compare the biological response to the four formulations. The inclusion criteria considered only sex, age, and weight, while discontinuation criteria included any situation affecting the animals’ welfare, such as intraoperative or postoperative complications.

## 3. Results and Discussion

### 3.1. Characterization of ZnO-NPs

The ZnO-NPs were synthesized according to previously reported methodologies [36]. The characterization of this component was described in previous work [34].

### 3.2. FT-IR of PCL/PLA/ZnO-NPs Blocks

Functional groups from PCL/PLA/ZnO-NPs blocks are shown in Figure 1. Symmetric vibrational modes exist at 3325 cm^−1^ from -OH and alkyl -CH groups at 2945 cm^−1^. Two peaks assigned to the vibrational mode of ester-like C=O groups present in the PLA and PCL components at 1756 and 1724 cm^−1^, the asymmetric vibrational mode of the C-O-C group corresponding to the aliphatic chain of the polymers at 1182 cm^−1^ and, finally, at 1085 cm^−1^ the stretching vibrational mode of the C-O group was observed. Concerning formulations F2, F3, and F4, the peak at 682 cm^−1^ attributed to the vibrational mode for the ZnO-NPs is not kept; however, it is noticeable that with the nanoparticle concentration increasing, the intensity of the CO group increases (1000–1182 cm^−1^), which might come from the PLA degradation (lactide, oligomeric rings, and acetaldehyde products) [42].

On the other hand, two vibrational modes at 1756 and 1724 cm^−1^ attributed to the carbonyl groups in F2 and F3 were observed, related to the PLA and PCL polymers [43,44]. For F4, only the peak at 1756 cm^−1^ was observed because the C=O groups interact with the ZnO-NPs promoting the accumulation of lactide molecules from the oligomeric ring and the acetaldehyde groups from the chain’s degradation induced by ZnO-NPs [42].

### 3.3. XRD of the PCL/PLA/ZnO-NP Blocks

Figure 2 shows the crystallographic characteristic peaks of PCL/PLA/ZnO-NPs. In the diffractogram, an orthorhombic PLA structure is assigned to the diffraction angles (2θ) 14.7, 16.6, 19.0, and 22.1, coinciding with previously published data [45]. Additionally, PCL diffraction peaks of orthorhombic nature were observed because it correlates with the same diffraction angle 2θ at 21.3 and 23.6, related to the (110) and (200) planes [46].

On the other hand, increasing the concentration of ZnO-NPs causes changes in the crystallinity in the F3 and F4 blocks, which probably suggests that the nanoparticles promote the loss of crystallinity when they are above 1% wt, decreasing the nucleation processes, giving greater mobility of the alkyl chains of the polymers with new diffraction sites, as observed in the diffraction angle 2θ at 9° [47]. Additionally, previous work has shown that introducing essential oils causes a decrease in the intensity of the diffraction peaks [34]. In this sense, it is observed that the intensity is preserved in the F1, F2, and F3 formulations, suggesting that the loss of crystallinity is more intense in F4 due to the generation of intermolecular spaces by the reduction in intramolecular interactions along the polymer chain [46].

### 3.4. Thermal Analysis of PCL/PLA/ZnO-NPs Blocks

Figure 3 shows the thermal behavior of the nanocomposites under heating. Within the thermograms, it was observed that as the concentration of ZnO-NPs increases, the thermal stability of the blocks decreases, which is probably related to the possibility of ZnO-NPs binding to the C=O groups to act as weak points in the polymer backbone. The introduction of ZnO-NPs makes them behave as crosslinking points between the polymer backbones, causing the formation of additional spaces between them, which decreases the strength of the intermolecular bond and therefore decreases their stability [48]. Furthermore, this poor thermal stability is also related to the lack of intermolecular interaction between the polymers, as mentioned in previous articles [49].

Figure 3A shows the thermograms of formulations F1, F2, F3, and F4. Regarding formulation F1 (30%PCL/70%PLA), two degradation stages are shown. The first degradation stage (between 285 and 340 °C) is associated with intramolecular cleavages leading to PLA degradation up to lactide, cyclic oligomers, acrylic acid, and acetaldehyde [50]. The second stage of degradation between 369 and 434 °C is attributed to the decomposition of the PCL caused by the cleavage of the hydroxyl group, generating depolymerization [9].

On the other hand, it can be observed that as the ZnO-NP concentration increases, the thermal stability of the blocks decreases as observed for the first stage of degradation, where for F2, it is approximately at 309 °C. In comparison, F4 presents the peak at 268 °C because samples containing ZnO-NPs intensify the degradation processes of the polymer matrix, leading to the production of low molecular weight PCL and PLA generated by intramolecular transesterification and depolymerization processes below 337 °C [51]. In other words, these degradative processes may be related to breaking the C=O and C-C bonds of the polymer matrix where ZnO-NPs act as crosslinking points inducing additional gaps between the polymers, decreasing the order promoted by the intermolecular bonds [52]. In other investigations [34], it has been observed that the introduction of plasticizing agents enhances thermal properties due to the presence of thermostable molecules, suggesting that it is necessary to introduce such molecules so that the thermal properties do not decay as the temperature increases [53,54].

Differential scanning calorimetry (DSC) allowed the elucidation of the thermal properties related to the formulations, as shown in Figure 4. The melting temperatures of PLA (T_m1_ and T_m2_) and PCL (T_m3_) are evident, as well as the crystallization temperature of PLA (T_cc_) and the glass transition temperature (T_g_). Table 2 summarizes the composite properties and the PLA crystallinity percentage.

The calculation of the crystallinity blocks becomes very complex due to the influence of ZnO-NPs since T_g_ for PLA (60 °C) is very close to the PCL melting point (T_m3_). Thermal characterization results reveal that as the concentration of ZnO-NPs is increased, a single packing of the PLA atoms is favored as it tends to show a melting temperature at 146 °C belonging to the α-form (pseudo-orthorhombic, pseudo-hexagonal, and orthorhombic) rather than its β-form (orthorhombic or trigonal). This observation suggests that the degradative action of ZnO-NPs favors the β-form packing [55]; furthermore, a low crystallinity PLA rate and degradation mediated by the ZnO-NPs stimulate the rise of more nucleation sites which, in turn, increase melting points for F2, F3, and F4 formulations [56,57].

The crystallization temperature was taken as the maximum value of the exothermic peak, where it was observed that the peak amplitude is interrupted as the ZnO-NP concentration increases until it disappears for F4, probably suggesting that ZnO-NPs delay the crystallization processes to a lesser extent for those not exceeding 1% wt (F2 and F3) while for 2% wt ZnO-NPs, it is more intense. This delay in crystallization may be due to the interaction between the carbonyl and hydroxyl groups of the polymers present and the ZnO-NPs, which hinders the macromolecular interaction, and, consequently, the formation of a stable crystalline structure [58,59].

### 3.5. Scanning Electron Microscopy (SEM) of PCL/PLA/ZnO-NP Blocks

The microstructure of the nanocomposites is shown in Figure 5. A porous structure favoring the interaction with other components is observed. In F1, the incompatibility between the polymers is shown with phase separation in the mixture, as followed in previous works [60]. Concerning formulations F2, F3, and F4, it is observed how the ZnO-NPs are deposited in the cavities of the blocks due to the change of contrast related to the conductivity of the material [61].

On the other hand, as can be observed, as the ZnO-NP concentration increases, the size and distribution of the cavities increase, probably due to the reduction in the elasticity of the PLA chains induced by the inadequate interfacial interactions between the nanoparticles and the polymeric matrix in the blocks, which could be the possible reason for the presence of voids causing new nucleation sites [62]. However, F4 presented a more compact structure due to a homogeneous distribution of the ZnO-NPs, which could elicitate the dehydration of the ZnO surfaces, decreasing the specific surface area [63].

### 3.6. In Vivo Biocompatibility Tests of the PCL/PLA/ZnO-NP Blocks

Figure 6 exhibits the implantation zone in the biomodels (dorsal area). Hair recovery without purulent exudate demonstrated a healthy recovery for all the biomodels. Subsequently, when staining analyses were carried out, the lesions caused by the material implantation were healed without an aggressive immune response. In addition, the presence of collagen fiber capsules encapsulating the material indicates that the healing process is underway since the collagen fibers allow the fixation of the material without an aggressive immune response until the removal for histological analysis.

#### 3.6.1. Biocompatibility of Formulation F1

F1 histological analysis after 60 days of subdermal implantation are observed in Figure 7. Several fragments are infiltrated by inflammatory cells without a fibrous capsule surrounding the implantation site (Figure 7A). However, using Masson’s trichrome staining, type I collagen fibers can be seen surrounding the large fragments (Figure 7B). At 100× magnification, particles are seen in contact with some inflammatory cells, immersed in connective tissue with blood vessels (Figure 7C).

In tissue engineering, scaffolds are used as structures that will enable cell growth and the formation of new tissue at sites of interest [64]. PLA is often used to produce scaffolds due to its biocompatibility, biodegradability, and osteoconductive properties. However, the biodegradability of PLA depends on the physicochemical properties imparted by the formed composite [64].

The PLA degradation/reabsorption process involves water molecules that attack hydro-labile bonds, breaking the polymer chain, releasing oligomers and monomers, causing the intervention of inflammatory cells, which will contribute to the complete disappearance of the material through enzymatic degradation and phagocytosis processes [64]; however, depending on the molecular weight and crystallinity, this process can be prolonged. Therefore, it is interesting to have a balance between degradation/reabsorption of the material and tissue neo-formation [64].

Polycaprolactone (PCL) is also biocompatible and biodegradable; when PCL is added to the PLA, a more subtle material is obtained, which retains the biocompatibility [65], but with a slower rate of absorption [66]. The F1 formulation consists of 30% PCL and 70% PLA, which promotes a healing process in which the host cells try to resorb the particles of the material. At 60 days of implantation, we observed that the tissue recovered its typical architecture and in the area of implantation numerous immersed material fragments in connective tissue with a few blood vessels persisted (Figure 7), indicating that, despite having used a medium molecular weight PLA, the PCL addition stabilizes the rate of degradation.

The histological results show that a fibrous capsule delimiting the implantation zone, reported for most implanted biomaterials, is not visible [67]. The finding of remnant material immersed in a connective tissue matrix with type I collagen indicates that a scarring process is taking place to recover the lost tissue in the implantation site.

#### 3.6.2. Biocompatibility of Formulation F2

Similar to that reported for F1, at 60 days after implantation, numerous fragments of F2 of variable size are observed (Figure 8A); however, a structure of parallel fibers delimiting the implantation zone was observed, compatible with a fibrous capsule (Figure 8A,B); the fragments of material are immersed in a connective tissue matrix with the presence of type I collagen fibers (Figure 8B), surrounded by inflammatory cells and with the presence of some blood vessels (Figure 8C); as the fragments of material are reduced to smaller sizes, they are surrounded by inflammatory cells for their phagocytosis process as shown in the green circle in Figure 8C.

Formulation F2 has 29.5% PCL, 70% PLA, and 0.5% ZnO-NPs. When comparing histological images of the F2 and F1 formulations, the only apparent difference appears to be the presence of the capsule (Figure 8A,B) and a slight increase in the number of inflammatory cells (Figure 7C and Figure 8C). This finding of the presence of a capsule surrounding the implanted material is similar to that reported by other authors who, for the same 60-day period, describe the existence of a capsule around PLA and PCL composites [68,69].

#### 3.6.3. Biocompatibility of Formulation F3

The results of the implantation of the F3 formulation show that fragments of variable size persist in the implantation zone, with the presence of a mild inflammatory infiltrate and numerous blood vessels (Figure 9A); the implantation zone is surrounded by a fragile type I collagen fiber capsule (blue fibers in Figure 9B). Also, in the implantation zone, numerous fragments of variable size are immersed in a connective tissue matrix with collagen fibers (Figure 9B). At 100× magnification, the pieces of the material are clearly in the process of degradation/resorption by inflammatory cells.

Histological images show an increase in the rate of degradation/resorption of the material supported by the observation of several small fragments and caused by the decrease in the percentage of PCL (29 wt.% in F3); an increase in the number of blood vessels is also observed (Figure 9A) which may have been caused by the rise in the percentage of ZnO-NPs (1%), in agreement with several reports that ZnO-NPs can stimulate vascular growth and angiogenesis, after promoting the formation of endothelial cells [70,71].

On the other hand, unlike the F1 and F2 formulations, when the PCL content was decreased, fibrous encapsulation was observed, which probably suggests a decrease in the degradation rate and the need for the organism to encapsulate the implanted material to control degradation; on the other hand, the incorporation of ZnO-NPs could have increased biocompatibility due to its effect on angiogenesis and healing, as reported in the literature [72,73].

#### 3.6.4. Biocompatibility of Formulation F4

The results of the F4 formulation implantations show that at 60 days, some medium-sized fragments of material and numerous small fragments persist immersed in a connective tissue matrix, and the implantation area is surrounded by a fibrillar structure compatible with a capsule (Figure 10A). At 40× magnification, the connective tissue matrix is composed of thin type I collagen fibers, inflammatory cells, and numerous blood vessels; collagen fibers surround the more significant type I particles according to Masson’s trichrome staining, but the smaller particles are not surrounded by fibers but by inflammatory cells, as can be seen in the area marked by a green circle in Figure 10B. Figure 4C, with a 100× magnification, exhibits inflammatory cells in contact with particles and several blood vessels.

Formulation F4 has PCL 28%, PLA 70%, and ZnO-NPs 2% as components; regarding the other three formulations, PLA has remained constant, PCL has decreased, and ZnO-NPs have increased. The function of PLA in this product was to provide the scaffold structure due to its mechanical characteristics; to prevent a speedy degradation, we incorporated PCL, which decreases the resorption rate while improving the mechanical properties and biocompatibility. We also included ZnO-NPs, which seem to have an effect in accelerating healing [74].

By decreasing the PCL percentage in the F4 formulation, a more significant degradation/reabsorption of the material was observed, manifested in the histological images as smaller particles and numerous microscopic particles in resorption. On the other hand, with the increase in ZnO-NPs, a more significant number of blood vessels was observed. Overall, the results show that by decreasing the PCL and increasing the ZnO-NPs, a biocompatible, biodegradable scaffold with angiogenic characteristics is obtained, with great potential in regenerative medicine.

The presence of the capsule may be related to the increased rate of resorption/degradation, as it has been reported that most biomaterials stimulate fibrous capsule formation during tissue healing [75]. In the early stages of the healing process, a temporary matrix is formed from plasma proteins; later, the provisional matrix is replaced by a connective tissue matrix composed of type III and type I collagen fibers, and as the healing process progresses and the grafted material is reabsorbed, this matrix is reabsorbed and replaced by a connective tissue similar to the original tissue at the implantation site [76].

In this research, the presence of the connective tissue matrix formed by type I collagen fibers was observed for all the formulations; however, in the results of the F4 formulation, the type I collagen fibers tend to disappear and are only observed surrounding the larger particles. In this case, better and faster healing occurred, explained by the decrease in PCL and the increase in the percentage of ZnO-NPs, which is considered angiogenic and elicit healing and biointegration [77,78].

## 4. Conclusions

In this study, we synthesized four PCL/PLA/ZnO-NP blocks, which exhibited improved thermal stability compared to their pure components. The incorporation of ZnO-NPs was confirmed by different characterization techniques such as FTIR, XRD, TGA, and DSC. In the FTIR, we observed the decrease in the band at 1725 cm^−1^ related to the C=O groups of the PCL, indicating that ZnO-NPs mainly interact with that moiety. In addition, we observed a decrease in the thermal properties and the crystallinity as the concentration of ZnO-NPs increased. The SEM analysis showed that the blocks have a porous morphology due to the homogeneous accumulation and dispersion between the polymeric chains facilitating a compact structure.

Furthermore, the PCL/PLA/ZnO-NPs blocks exhibited preliminary biocompatibility after a degradation/resorption and a fast-healing process, with a complete recovery of tissue architecture. Notably, there was no aggressive immune response observed. During the resolution processes, the material was fragmented and engulfed by the inflammatory cells, stimulating the proliferation of collagen fibers, blood vessels, and cells, continuing the process of material resorption, and demonstrating a preliminary biocompatible process. The results reported here demonstrate the relevance and potential impact of the ZnO-NP-based scaffolds in soft tissue regeneration. 

## Figures and Tables

**Figure 1 pharmaceutics-15-02196-f001:**
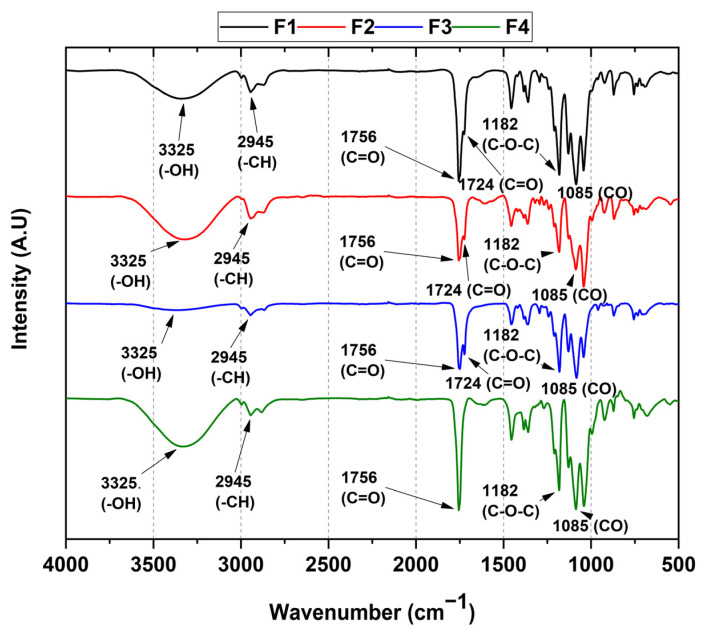
FT-IR spectrum of PCL/PLA/ZnO-NP blocks. F1, 30%PCL/70%PLA; F2, 29.5%PCL/70%PLA/0.5%ZnO-NPs; F3, 29%PCL/70%PLA/1%ZnO-NPs; F4, 28%PCL/70%PLA/2%ZnO-NPs.

**Figure 2 pharmaceutics-15-02196-f002:**
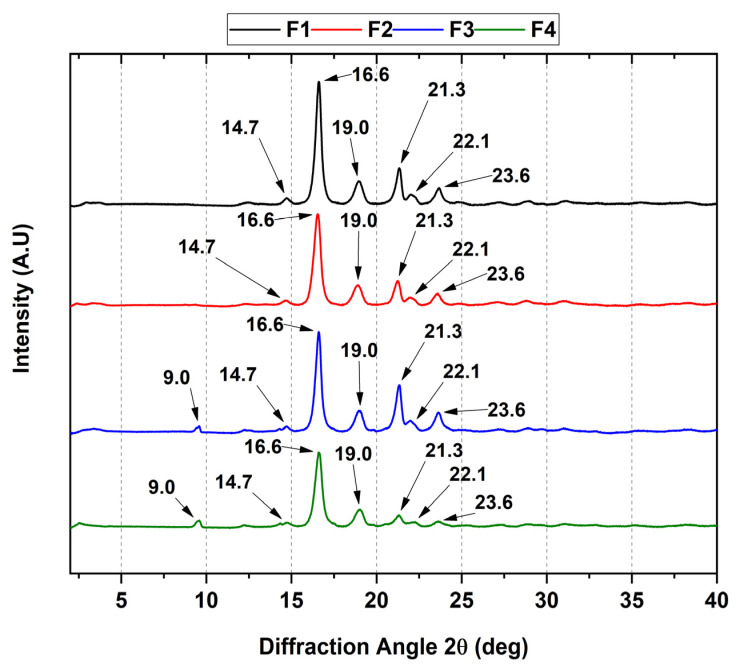
XRD analysis of PCL/PLA/ZnO-NPs blocks. F1, 30%PCL/70%PLA; F2, 29.5%PCL/70%PLA/0.5%ZnO-NPs; F3, 29%PCL/70%PLA/1%ZnO-NPs; F4, 28%PCL/70%PLA/2%ZnO-NPs.

**Figure 3 pharmaceutics-15-02196-f003:**
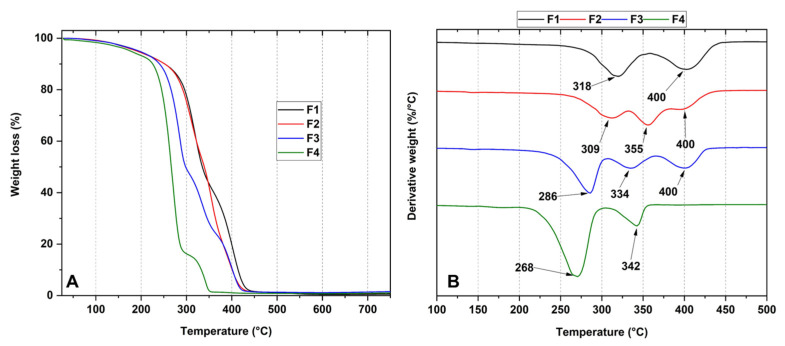
Thermogram (**A**) and its derivative curves (**B**) of the PCL/PLA/ZnO-NPs blocks. F1, 30%PCL/70%PLA; F2, 29.5%PCL/70%PLA/0.5%ZnO-NPs; F3, 29%PCL/70%PLA/1%ZnO-NPs; F4, 28%PCL/70%PLA/2%ZnO-NPs.

**Figure 4 pharmaceutics-15-02196-f004:**
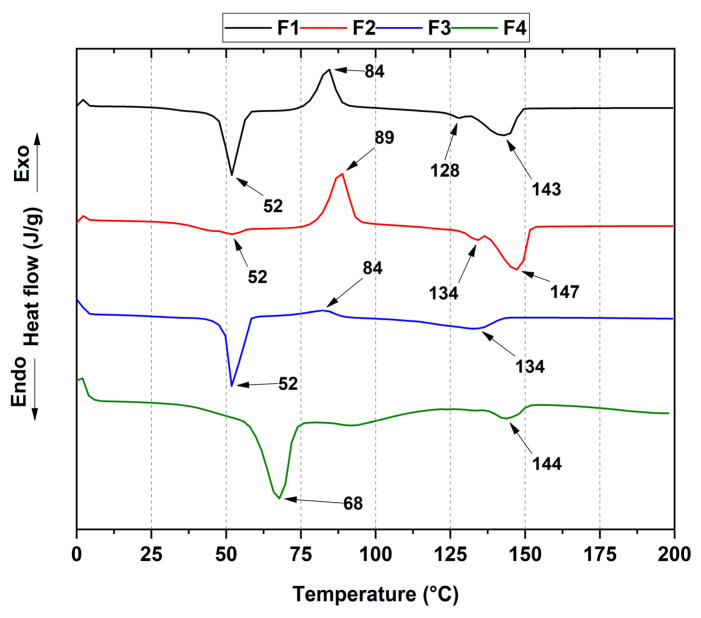
DSC curves of the PCL/PLA/ZnO-NPs nanocomposites. F1, 30%PCL/70%PLA; F2, 29.5%PCL/70%PLA/0.5%ZnO-NPs; F3, 29%PCL/70%PLA/1%ZnO-NPs; F4, 28%PCL/70%PLA/2%ZnO-NPs.

**Figure 5 pharmaceutics-15-02196-f005:**
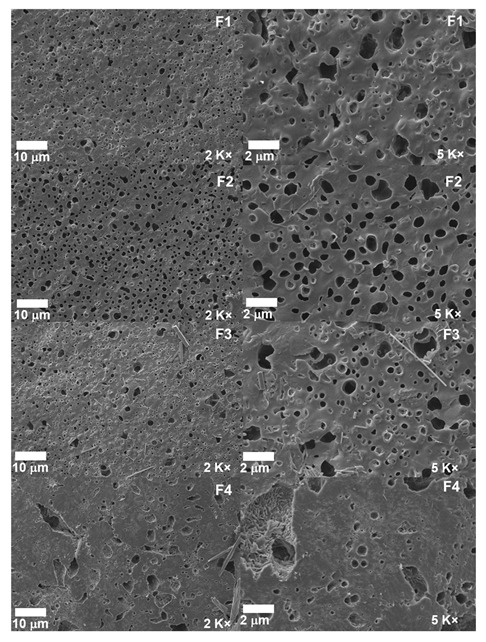
Morphology of PCL/PLA/ZnO-NPs blocks by SEM. F1, 30%PCL/70%PLA at 2000 and 5000×; F2, 29.5%PCL/70%PLA/0.5%ZnO-NPs at 2000 and 5000×; F3, 29%PCL/70%PLA/1% ZnO-NPs at 2000 and 5000×; F4, 28%PCL/70%PLA/2%ZnONPs at 2000 and 5000×.

**Figure 6 pharmaceutics-15-02196-f006:**
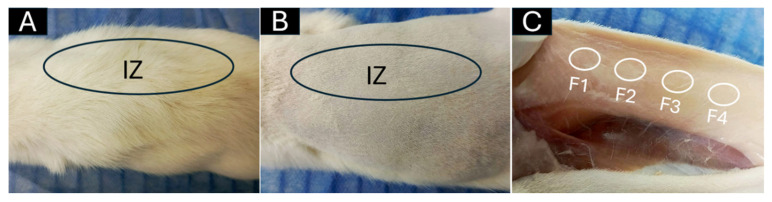
Subdermal dorsal implantation zone. (**A**) Dorsal area with abundant hair. (**B**) Trichotomy of dorsal area. (**C**) Subdermal implantation area. Black ovals: implantation zone. White circles: blocks implanted. F1–F4: formulations 1, 2, 3, and 4. IZ: Implantation zone.

**Figure 7 pharmaceutics-15-02196-f007:**
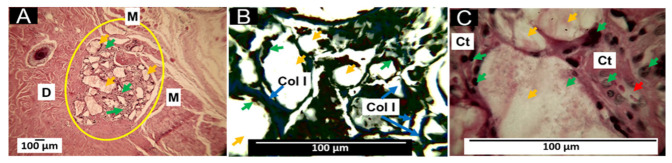
Subdermal implantations F1 at 60 days. (**A**) F1 formulation at 4× HE technique. (**B**) F1 formulation at 40× MT technique. (**C**) F1 formulation at 100× HE technique. D: dermis. M: muscle. Yellow oval: implantation zone. Yellow arrows: fragments of F1 material; green arrows: inflammatory cells; blue arrows: collagen type I fibers. Col I: type I collagen—Ct: connective tissue; red arrow: blood vessel.

**Figure 8 pharmaceutics-15-02196-f008:**
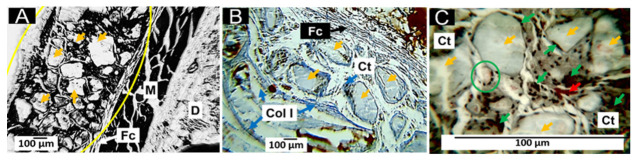
Subdermal implantations of F2 at 60 days. (**A**) F2 formulation at 10× HE technique. (**B**) F2 formulation at 10× MT technique. (**C**) F2 formulation at 100× HE technique. D: dermis. M: muscle. Yellow oval: implantation zone. Fc: fibrous capsule. Yellow arrows: fragments of material F2; blue arrows: type I collagen fibers. Green arrows: inflammatory cells. Red arrows: blood vessel. Col I: type I collagen. Ct: connective tissue. Green circle: area of resorption by cells.

**Figure 9 pharmaceutics-15-02196-f009:**
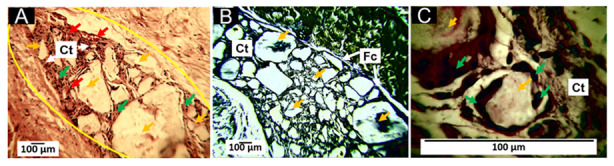
Subdermal implantations of F3 at 60 days. (**A**): F3 formulation at 10× HE technique. (**B**): F3 formulation at 10× MT technique. (**C**): F3 formulation at 100× HE technique. Yellow oval: implantation zone. Ct: connective tissue. Yellow arrows: F3 material fragments. Green arrows: inflammatory cells. Red arrows: blood vessels. Fc: fibrous capsule.

**Figure 10 pharmaceutics-15-02196-f010:**
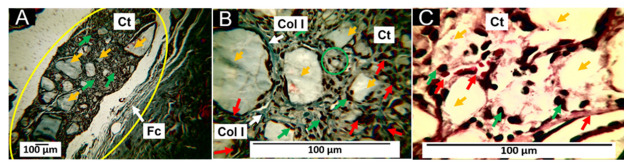
Subdermal implantations of F4 at 60 days. (**A**): Formulation F4 at 10× MT technique. (**B**): Formulation F4 at 40× MT technique. (**C**): Formulation F4 at 100× HE technique. Yellow oval: implantation zone. Ct: connective tissue. Yellow arrows: fragments of material F3. Green arrows: inflammatory cells. Fc: fibrous capsule. Red arrows: blood vessels. Col I: collagen type I. Green circle: zone of reabsorption by cells.

**Table 1 pharmaceutics-15-02196-t001:** Formulation of PCL/PLA/ZnO-NP blocks.

Components	PCL (%)	PLA (%)	ZnO-NPs (%)
F1	30	70	0
F2	29.5	70	0.5
F3	29	70	1
F4	28	70	2

**Table 2 pharmaceutics-15-02196-t002:** Thermal properties of blocks PCL/PLA/ZnO-NPs. F1, 30%PCL/70%PLA; F2, 29.5%PCL/70%PLA/0.5%ZnO-NPs; F3, 29%PCL/70%PLA/1%ZnO-NPs; F4, 28%PCL/70%PLA/2%ZnO-NPs.

	T_g_ (°C)	T_cc_ (°C)	T_m1_ (°C)	T_m2_ (°C)	T_m3_ (°C)	Xc PLA (%)
F1	55	84	128	143	52	5.1
F2	57	89	134	147	52	4.9
F3	55	84	134	-	52	4.7
F4	52	-	-	144	68	-

## Data Availability

Data will be available on request from the corresponding author.

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
