# Peer review of "Biocompatibility Assessment of Polycaprolactone/Polylactic Acid/Zinc Oxide Nanoparticle Composites Under In Vivo Conditions for Biomedical Applications"

_pharmaceutics, 2023, doi:10.3390/pharmaceutics15092196_

Round 1
Reviewer 1 Report
The article "Polylactic acid (PLA)/Polycaprolactone (PCL)/Zinc oxide nanoparticles (ZnO-NPs) composites and their in vivo biocompatibility studies" describes the synthesis and characterization of some ZnO-based composites used as biomaterials in subdermal applications. It is a valuable study that can be published after authors address the following problems:
Evaluate if the abbreviations are required in title. They must be explained at first use in Abstract and text body. If “zinc oxide” form is used in title, then use “ZnO” form as keyword. I suggest use biomaterial as keyword too.
I suggest keeping the wording order across the manuscript. As composite is labeled PCL/PLA/ZnO, means the title order should be Polycaprolactone/ Polylactic acid / Zinc oxide...and in abstract at row 25 PLA/PCL should be PCL/PLA; at row 169 is again PLA/PCL/ZnO....
Abstract should be checked and revised carefully by briefly introducing the work plan and key findings. Abstracts should highlight the innovation of the article, as often abstract section is presented separately in search engines, it must be able to stand alone as an informative piece. In the abstract, authors need to focus more on the quantitative information, not qualitative one. Some English terms must be rethink for the clarity sake like “ZnO-NPs degradation excels over thermal properties....”
Explanations like “the morphology of the blocks with 2 wt. % of ZnO-NPs exhibited a homogeneous morphology, possibly due to OH group removal on the ZnO-NPs periphery” must be sustained with clear evidences. Why authors think the –OH moieties will not be removed at 0.5% and 1% ZnO concentration?
In introduction section please clearly state the hypothesis. Which exact problem was supposed to be solved by the present research?
Zinc is a key element in matrix metalloproteinases, they being responsible for the extracellular matrix, and are directly related to the tissue development and angiogenesis. Using ZnO-composites in vivo can promote the activity of these endopeptidases as zinc ions are released from the ZnO NPs (see doi: 10.3390/ijms21249739; doi: 10.3390/ijms23031806).
The addition of ZnO NPs into a polymeric mixture can modify the crystallinity as the NPs can act as crosslinking points between polymer backbones and induces additional spaces between them, decreasing the order promoted by intermolecular bonds (introduction, XRD and TG section). The authors can use doi: 10.3390/pharmaceutics13071020 as discussing starting point for this feature and can also use it as a reference for ZnO-based composite used as food packaging.
Authors should better underline the importance of ZnO NPs in composite biomaterials. The most important aspects related to ZnO NPs should be clearly presented in order to provide a properly description of the state of art in this field. Under introduction, at rows 75-78 please discuss more and update with doi: 10.3390/pharmaceutics14122842 for the antimicrobial activity of ZnO NPs relating to the size and morphology of the nanoparticles and also the mechanism of nanoparticle dissolution and release of Zn2+ ions.
In section 2.4. please indicate which peak was used for ZnO crystallite size calculation. The XRD of ZnO is not presented in manuscript, therefore please indicate that characterization was reported in [24]. For FTIR spectra please indicate the use of ATR module, resolution of measurements, averaging of how many spectra etc (like in section 2.5).
In section 3.2 FTIR, please use the established terminology. Instead of “tension” bands use peaks assigned to vibration or deformation modes. In general, the English language needs some polishing for style and typos (e.g. row 122 “each block's four weight percent”; use proper indices in formula like row 110 ZnCl2; at rows 173-175 sentence has missing words: For determination of the glass transition....; row 378 Nps;
Authors have used composites with larger amount of PLA vs PCL. Both polymers have C=O bonds. PLA has twice the number. The literature indicate that PLA present the vibration mode at ~1750 cm-1 (doi: 10.3390/polym6010093) while PCL has it at ~1720-1730 cm-1 (doi: 10.3390/polym12030717). It is safe to assume that C=O moieties from PCL are the main groups interacting with ZnO NPs, as this peak is disappearing in 2% ZnO sample, while the 1756 cm-1 remains. Authors can additionally indicate the steric impediments from PLA for this preferential interaction with C=O moiety of PCL.
In XRD section please give the ASTM / JCPDS card for PLA and PCL. For PLA this reviewer cannot find relevant literature that indicates those Miller indices.
For TG section “degradative processes involved in ZnO-NPs” is not a suitable explanation. ZnO is exceptionally stable from thermal point of view, with no mass loss up to 900oC (only surface –OH moieties being eliminated). As thermal stability decrease for the composites with higher % of ZnO I suggest authors to explore the possibility that ZnO bond to C=O moieties, as discussed before in FTIR, to act as weak points in polymer backbone from where fragmentation can easily occur. Authors will find similar findings in above suggested literature (doi: 10.3390/pharmaceutics13071020). In addition, as suggested for introduction and XRD discussion, introducing ZnO NPs that act as crosslinking points between polymer backbones and induces additional spaces between them, decreasing the strength of intermolecular bonds, therefore decreasing the stability.
Authors must explain why in Figure 3 all samples have a residual mass of 0. The working conditions were declared as inert atmosphere (N2), therefore a carbonaceous residue + ZnO should have been presented. Additionally, please label with A) and B) the two images in Figure 3 and update the description.
Again English polishing at row 319 “F4 is kept at 268°C” should be “F4 presents the peak at 268oC” or something similar.
Who is TTEO from Figure 4 caption?
There is no statistical analysis in biocompatibility part.
The conclusion should reflect the heuristic of the study. How is this system a better one? Conclusion section must be reworked to underline the novelty and advantages of this research, with actual numbers.
In general, the English language needs some polishing for style and typos (e.g. row 122 “each block's four weight percent”; use proper indices in formula like row 110 ZnCl2; at rows 173-175 sentence has missing words: For determination of the glass transition....; row 378 Nps.
In section 3.2 FTIR, please use the established terminology. Instead of “tension” bands use peaks assigned to vibration or deformation modes.
Author Response
We appreciate all the comments from the reviewer. All the answers are addressed point by point for clarity.
Reviewer 1
The article "Polylactic acid (PLA)/Polycaprolactone (PCL)/Zinc oxide nanoparticles (ZnO-NPs) composites and their in vivo biocompatibility studies" describes the synthesis and characterization of some ZnO-based composites used as biomaterials in subdermal applications. It is a valuable study that can be published after the authors address the following problems:
Evaluate if the abbreviations are required in title. They must be explained at first use in Abstract and text body. If "zinc oxide" form is used in title, then use "ZnO" form as keyword. I suggest use biomaterial as keyword too.
R// Thank you very much for your suggestion. We eliminated abbreviations in the title, and the keywords were introduced according to your proposal.
I suggest keeping the wording order across the manuscript. As composite is labeled PCL/PLA/ZnO, means the title order should be Polycaprolactone/ Polylactic acid / Zinc oxide...and in Abstract at row 25 PLA/PCL should be PCL/PLA; at row 169 is again PLA/PCL/ZnO....
R// Thank you very much for the observation. We change the order of the components according to the suggestion, setting PCL as the initial acronym.
Abstract should be checked and revised carefully by briefly introducing the work plan and key findings. Abstracts should highlight the innovation of the article, as often abstract section is presented separately in search engines, it must be able to stand alone as an informative piece. In the Abstract, authors need to focus more on the quantitative information, not qualitative one. Some English terms must be rethink for the clarity sake like "ZnO-NPs degradation excels over thermal properties...."
R// We appreciate your suggestion. The new Abstract is shown in the following paragraph:
The increasing demand for non-invasive biocompatible materials in biomedical applications, driven by accidents and diseases like cancer, has led to the development of sustainable biomaterials. Here, we report the synthesis of four block formulations using polycaprolactone (PCL), polylactic acid (PLA), and zinc oxide nanoparticles (ZnO-NPs) for subdermal tissue regeneration. Characterization by Fourier transform infrared spectroscopy (FT-IR) and X-ray diffraction (XRD) confirmed the composition of the composites. Additionally, the interaction of ZnO-NPs mainly occurred with the C=O groups of PCL occurred at 1724cm-1, which disappears for F4, as evidenced in the FT-IR analysis. Likewise, this interaction evidenced the decrease of the crystallinity of the composites as they act as crosslinking points between the polymer backbones, inducing gaps between them and weakening the strength of the intermolecular bonds. Thermogravimetric (TGA) and Differential Scanning Calorimetry (DSC) analyses confirmed that the ZnO-NPs bind to the carbonyl groups of the polymer, acting as weak points in the polymer backbone from where the different fragmentations occur. Scanning electron microscopy (SEM) showed that the increase in ZnO-NPs facilitated a more compact surface due to the excellent dispersion and homogeneous accumulation between the polymeric chains, facilitating this morphology. The in vivo studies using the nanocomposites demonstrated a degradation/resorption of the blocks in a ZnO-NPs-depending mode. The degradation mechanism involved the fragmentation of larger particles and the presence of connective tissue comprising type I collagen fibers, blood vessels, and inflammatory cells, continuing the resorption of the implanted material. The results reported here demonstrate the relevance and potential impact of the scaffolds in soft tissue regeneration.
Explanations like "the morphology of the blocks with 2 wt. % of ZnO-NPs exhibited a homogeneous morphology, possibly due to OH group removal on the ZnO-NPs periphery" must be sustained with clear evidences. Why authors think the –OH moieties will not be removed at 0.5% and 1% ZnO concentration?
R// Thank you very much for the suggestion. The following paragraph was introduced in the Abstract between lines 30-33 to avoid confusion on how ZnO-Nps act to give a compact surface when the highest concentration of this material is found,
Scanning electron microscopy (SEM) showed that the increase in ZnO-NPs facilitated a more compact surface due to the excellent dispersion and homogeneous accumulation between the polymeric chains, introducing more interaction between the polymeric chains and the NPs, and producing a homogeneous morphology
In introduction section please clearly state the hypothesis. Which exact problem was supposed to be solved by the present research?
R// Thank you very much for the suggestion. You can find the correction between Lines 111-119.
However, no PCL/PLA/ZnO-NPs scaffolds have been studied in animal models as a first approximation to current medical devices. In this work, we propose four formulations of PCL/PLA/ZnO-NPs blocks to be evaluated subdermally after 60 days of implantation in Wistar rats to observe the material's biocompatibility. The biocompatibility results showed the presence of connective tissue constituted mainly by type I collagen for all the formulations; nevertheless, for F4, a smaller arrangement of these type I fibers was observed, which is related to an improvement in the healing process probably due to the increase in the concentration of ZnO-NPs as well as to the decrease of PCL.
Zinc is a key element in matrix metalloproteinases, they being responsible for the extracellular matrix, and are directly related to the tissue development and angiogenesis. Using ZnO-composites in vivo can promote the activity of these endopeptidases as zinc ions are released from the ZnO NPs (see doi: 10.3390/ijms21249739; doi: 10.3390/ijms23031806).
Thank you very much for your comment. The information was added as follows (lines 93-96):
It has also been studied in the development of Zn-dependent matrix metalloproteinases due to its ability to degrade almost all components of the extracellular matrix responsible for cellular repair and tissue remodeling.
The addition of ZnO NPs into a polymeric mixture can modify the crystallinity as the NPs can act as crosslinking points between polymer backbones and induces additional spaces between them, decreasing the order promoted by intermolecular bonds (introduction, XRD and TG section). The authors can use doi: 10.3390/pharmaceutics13071020 as discussing starting point for this feature and can also use it as a reference for ZnO-based composite used as food packaging.
R// Thank you very much for the suggestion. Indeed, adding ZnO-NPs generates intermolecular spaces between the polymers used due to their degradative properties. In this sense, concerning XRD, we added the following text (lines 324-327):
In this sense, it is observed that the intensity is preserved in F1, F2, and F3 formulations, suggesting that the crystallinity loss is more intense in F4 due to the generation of intermolecular spaces by the reduction of intramolecular interactions along the polymer chain.
On the other hand, following the suggestion, we added the following text in the TG section (lines 356-360):
In other words, these degradative processes may be related to the breaking of the C=O and C-C bonds of the polymer matrix, where ZnO-NPs act as crosslinking points inducing additional gaps between the polymers, decreasing the order promoted by the intermolecular bonds.
Authors should better underline the importance of ZnO NPs in composite biomaterials. The most important aspects related to ZnO NPs should be clearly presented in order to provide a properly description of the state of art in this field. Under introduction, at rows 75-78 please discuss more and update with doi: 10.3390/pharmaceutics14122842 for the antimicrobial activity of ZnO NPs relating to the size and morphology of the nanoparticles and also the mechanism of nanoparticle dissolution and release of Zn2+ ions
R// Thank you very much for the suggestion. Details can be found between Lines 77-85 and lines 88-98:
ZnO-NPs are synthesized by precipitation [1], sol-gel method [2], spray pyrolysis [3], thermal decomposition [4], and forced solvolysis [5]. Each method has both advantages and disadvantages. Additional methods have been reported, such as calcination to prepare ZnO-NPs from the precipitation method [6], disposition of raw materials from the natural extracts method, nanodomain overlay by the spray pyrolysis method [7], and production of zinc acetate by the forced solvolysis method [8]. However, the latter has allowed morphological control depending on the type of alcohol and the number of carbons present. For example, methanol resulted in nanospheres, while polyhedral morphology was obtained when using 1-butanol and becoming rod-shaped for 1-hexanol [5].
The mechanism of the antimicrobial activity of ZnO-NPs is not described. However, there are two separate pathways [5,9]. In the presence of light, ZnO-NPs produce reactive oxygen species (ROS) responsible for the oxidative stress produced in the microorganisms across the cell membrane. On the other hand, under dark conditions, ZnO-NPs mechanically pierce the cell membrane and interfere with the essential constituents of the microbial cell. It has also been studied in the development of Zn-dependent matrix metalloproteinases due to its ability to degrade almost all components of the extracellular matrix responsible for cellular repair and tissue remodeling [10]. Due to their potent antimicrobial activity, ZnO-NPs have been studied in topical ointments [11], controlled-release drugs [12], tissue dressings [13], and as components in antimicrobial packaging [14].
In section 2.4. please indicate which peak was used for ZnO crystallite size calculation. The XRD of ZnO is not presented in manuscript, therefore please indicate that characterization was reported in [24]. For FTIR spectra please indicate the use of ATR module, resolution of measurements, averaging of how many spectra etc (like in section 2.5).
R// Thank you very much for the observation. The details on how the crystallite size calculation and measurements were done were added between lines 153-154 and 162-164:
"Using the attenuated total reflectance (ATR) method with a diamond tip in the transmittance mode and a spectral resolution of 4 cm-1 at 32 scans."
The average crystallite size of the ZnO-NPs was determined using the diffraction peaks (2θ) 32, 34, and 36 attributed to the (100), (002), and (101) planes.
In section 3.2 FTIR, please use the established terminology. Instead of "tension" bands use peaks assigned to vibration or deformation modes. In general, the English language needs some polishing for style and typos (e.g. row 122 "each block's four weight percent"; use proper indices in formula like row 110 ZnCl2; at rows 173-175 sentence has missing words: For determination of the glass transition....; row 378 Nps;
R// Thank you very much for the suggestion. The terms presented in the FTIR discussion have already been changed.
Authors have used composites with larger amount of PLA vs PCL. Both polymers have C=O bonds. PLA has twice the number. The literature indicate that PLA present the vibration mode at ~1750 cm-1 (doi: 10.3390/polym6010093) while PCL has it at ~1720-1730 cm-1 (doi: 10.3390/polym12030717). It is safe to assume that C=O moieties from PCL are the main groups interacting with ZnO NPs, as this peak is disappearing in 2% ZnO sample, while the 1756 cm-1 remains. Authors can additionally indicate the steric impediments from PLA for this preferential interaction with C=O moiety of PCL.
R// Thank you very much for the observation. The manuscript was modified between lines 294-298.
On the other hand, two vibrational modes at 1756 and 1724 cm-1 attributed to the carbonyl groups in F2 and F3 were observed, related to the PLA and PCL polymers [15,16]. For F4, only the peak associated with PLA at 1756 cm-1 was observed, while the peak at 1724 cm-1, indicating that it is the central peak interacting with ZnO-NPs. The C=O groups interact with the ZnO-NPs promoting the accumulation of lactide molecules from the oligomeric ring and the acetaldehyde groups from the chain's degradation induced by ZnO-NPs [17].
In XRD section please give the ASTM / JCPDS card for PLA and PCL. For PLA this reviewer cannot find relevant literature that indicates those Miller indices.
R// Thank you very much for your suggestion. According to the literature, there is no convergence of the characteristic planes of PLA, while its crystallographic chart is not found due to the polymorphism of the α or β form. Therefore, we decided to eliminate the planes for PLA and show the diffractogram without the planes for each crystallographic peak (Lines 294-296).
For TG section "degradative processes involved in ZnO-NPs" is not a suitable explanation. ZnO is exceptionally stable from thermal point of view, with no mass loss up to 900°C (only surface –OH moieties being eliminated). As thermal stability decrease for the composites with higher % of ZnO I suggest authors to explore the possibility that ZnO bond to C=O moieties, as discussed before in FTIR, to act as weak points in polymer backbone from where fragmentation can easily occur. Authors will find similar findings in above suggested literature (doi: 10.3390/pharmaceutics13071020). In addition, as suggested for introduction and XRD discussion, introducing ZnO NPs that act as crosslinking points between polymer backbones and induces additional spaces between them, decreasing the strength of intermolecular bonds, therefore decreasing the stability.
R// Thank you very much for the observation. Regarding the thermal stability of the composites, the text was modified between lines 333-337:
"…which is probably related to the possibility of ZnO-NPs binding to the C=O groups acting as weakening points in the polymer backbone. The introduction of ZnO-NPs introduces crosslinking between the polymer backbones, causing the formation of additional spaces between them, decreasing the strength of the intermolecular bond and, therefore, their stability. "
Authors must explain why in Figure 3 all samples have a residual mass of 0. The working conditions were declared as an inert atmosphere (N2). Therefore a carbonaceous residue + ZnO should have been presented. Additionally, please label A) and B) the two images in Figure 3 and update the description.
R// Thank you very much for your appreciation. In a similar work where the degradation of PLA with ZnO-NPs is evaluated by thermogravimetric analysis using nitrogen, as shown in Figure 1, residual masses higher than 5% are not observed until the concentration of ZnO-NPs exceeds 10 %wt. In the present work, something similar is observed, probably due to the interaction between the ZnO-NPs and the carbonyl groups, which facilitates the degradation of the whole carbon skeleton. In our study, a residual mass of 1% is generally observed, probably related to the concentration of ZnO-NPs present in each formulation.
Figure 1. Image taken from the reference [18]. Thermogravimetric analysis of PLA and PLA-based electrospun nanofibers (a) weight vs. temperature and (b) derivative curves.
Again, English polishing at row 319 "F4 is kept at 268°C" should be "F4 presents the peak at 268oC" or something similar.
R// Thank you very much for the suggestion. We rewrote the text between Lines 341-344:
F4 presents the peak at 268°C because samples containing ZnO-NPs intensify the degradation processes of the polymer matrix, leading to the production of low molecular weight PCL and PLA generated by intramolecular transesterification and depolymerization processes below 337°C.
Who is TTEO from Figure 4 caption?
R// Thank you very much for the suggestion. It was a mistake. The expression was changed as follows:
DSC curves of the PCL/PLA/ZnO-NPs nanocomposites
There is no statistical analysis in biocompatibility part.
R// Thank you very much for the suggestion. In this section, we did not perform statistical analysis because we were not determining a variable over time. Instead, a qualitative analysis of the implantation site was completed to observe whether a necrotic process and the presence of blood vessels, connective tissue, and collagen fibers, among other systems, were found.
The conclusion should reflect the heuristic of the study. How is this system a better one? Conclusion section must be reworked to underline the novelty and advantages of this research, with actual numbers.
R// We appreciate the suggestion. In the text, the conclusion can be found as follows:
In this study, we synthesized four PCL/PLA/ZnO-NPs blocks, which exhibited improved thermal stability compared to their pure components. The incorporation of ZnO-NPs was confirmed by different characterization techniques such as FTIR, XRD, TGA, and DSC. In the FTIR, we observed the decrease of the band at 1725 cm-1 related to the C=O groups of the PCL, indicating that ZnO-NPs mainly interact with that moiety. In addition, we observed a decrease in the thermal properties and the crystallinity as the concentration of ZnO-NPs increased. The SEM analysis showed that the blocks have a porous morphology due to the homogeneous accumulation and dispersion between the polymeric chains facilitating a compact structure.
Furthermore, the PCL/PLA/ZnO-NPs blocks exhibited preliminary biocompatibility after a degradation/resorption and a fast-healing process, with a complete recover tissue architecture. Notably, there was no aggressive immune response observed. During the resolution processes, the material was fragmented and engulfed by the inflammatory cells, stimulating the proliferation of collagen fibers, blood vessels, and cells, continuing the process of material resorption, and demonstrating a preliminary biocompatible process. The results reported here demonstrate the relevance and potential impact of the ZnO-NPs-based scaffolds in soft tissue regeneration.
In general, the English language needs some polishing for style and typos (e.g. row 122 "each block's four weight percent"; use proper indices in formula like row 110 ZnCl2; at rows 173-175 sentence has missing words: For determination of the glass transition....; row 378 Nps.In section 3.2 FTIR, please use the established terminology. Instead of "tension" bands use peaks assigned to vibration or deformation modes.
R// Thank you very much for the grammatical remarks. The document was amended following the reviewer's suggestions.

Reviewer 2 Report
Regarding the manuscript (pharmaceutics-2566758) entitled:
“Polylactic acid (PLA)/Polycaprolactone (PCL)/Zinc oxide nanoparticles (ZnO-NPs) composites and their in vivo biocompatibility studies”
Comments to the Author
General comment
The manuscript describes development of polycaprolactone (PCL), polylactic acid (PLA), and zinc oxide nanoparticles (ZnO-NPs) for subdermal tissue regeneration. I have some few comments to be considered before publication:
1. Title: It is not clear the use of this composite
2. Abstract: Please data and number in the abstract to provide some insight into the results. Also, at the beginning you mentioned “diseases like cancer” then you mentioned “for subdermal tissue regeneration”. This will confuse the reader about the objective of the study. What is the meaning of four block formulations?
3. Introduction: should be enriched with studies using the composite.
4. Table 2. Formulation of PCL/PLA/ZnO-NPs blocks: Please make the name of component as columns and formulations as rows.
5. FT-IR of PCL/PLA/ZnO-NPs blocks: Why the -OH group disappeared in F3?
6. DSC curves of the PCL/PLA/ZnO-NPs/TTEO nanocomposites. Why did the Tg point at 52 disappear for F2?
7. Figure 5. Morphology of PCL/PLA/ZnO-NPs blocks by SEM: More images with higher magnification should be added to observe the NPs.
8. Figure 8. Subdermal implantations of F2 at 60 days: Images are not colored.
Author Response
We appreciate all the comments from the reviewer. All the answers are addressed point by point for clarity.
Reviewer 2
General comment
The manuscript describes development of polycaprolactone (PCL), polylactic acid (PLA), and zinc oxide nanoparticles (ZnO-NPs) for subdermal tissue regeneration. I have some few comments to be considered before publication:
- Title: It is not clear the use of this composite
R// Thank you very much for the comment. The title was changed as follows.
Biocompatibility Assessment of Polycaprolactone /Polylactic acid /Zinc oxide nanoparticles composites Under In Vivo Conditions for Biomedical Applications
- Abstract: Please data and number in the Abstract to provide some insight into the results. Also, at the beginning you mentioned "diseases like cancer" then you mentioned "for subdermal tissue regeneration". This will confuse the reader about the objective of the study. What is the meaning of four block formulations?
R// We appreciate your suggestion. The Abstract was rewritten as follows:
The increasing demand for non-invasive biocompatible materials in biomedical applications, driven by accidents and diseases like cancer, has led to the development of sustainable biomaterials. Here, we report the synthesis of four block formulations using polycaprolactone (PCL), polylactic acid (PLA), and zinc oxide nanoparticles (ZnO-NPs) for subdermal tissue regeneration. Characterization by Fourier transform infrared spectroscopy (FT-IR) and X-ray diffraction (XRD) confirmed the composition of the composite. Additionally, the interaction of ZnO-NPs mainly occurred with the C=O groups of PCL occurred at 1724cm-1, which disappears for F4, as evidenced in the FT-IR analysis. Likewise, this interaction evidenced the decrease of the crystallinity of the composites as they act as crosslinking points between the polymer backbones, inducing gaps between them and weakening the strength of the intermolecular bonds. Thermogravimetric (TGA) and Differential Scanning Calorimetry (DSC) analyses confirmed that the ZnO-NPs bind to the carbonyl groups of the polymer, acting as weak points in the polymer backbone from where the different fragmentations occur. Scanning electron microscopy (SEM) showed that the increase in ZnO-NPs facilitated a more compact surface due to the excellent dispersion and homogeneous accumulation between the polymeric chains, facilitating this morphology. The in vivo studies using the nanocomposites demonstrated a degradation/resorption of the blocks in a ZnO-NPs-depending mode. The degradation mechanism involved the fragmentation of larger particles and the presence of connective tissue comprising type I collagen fibers, blood vessels, and inflammatory cells, continuing the resorption of the implanted material. The results reported here demonstrate the relevance and potential impact of the scaffolds in soft tissue regeneration.
- Introduction: should be enriched with studies using the composite.
R// Thank you very much for the suggestion. We added some references in the introduction section, including the three components of the mixture, for therapeutic applications and the use of this polymeric matrix for the controlled release of ZnO-NPs. Additionally, the present work extended the results of previous work, where tea tree essential oil was added, and the effect on biomodels was evaluated. In this sense, we consider the impact of using a similar composite to the one used previously but increasing the concentration of ZnO-NPs and removing the essential oil to understand the effect of ZnO-NPs on the scaffold's properties.
- Table 1. Formulation of PCL/PLA/ZnO-NPs blocks: Please make the name of component as columns and formulations as rows.
R// Thank you very much for your comment. Table 1 was modified as follows:
- FT-IR of PCL/PLA/ZnO-NPs blocks: Why the -OH group disappeared in F3?
R// We appreciate your suggestion. Concerning formulation F3, the -OH band present in the composite (figure 1) decreases in its intensity, probably due to the interaction between the ZnO-NPs and the carbonyl groups of the PCL, which promotes the destabilization of the polymeric matrix generating acetaldehyde and the accumulation of lactide molecules from the oligomeric rings. Interestingly, once the concentration of ZnO-NPs is increased, the C=O vibrational mode attributed to PCL is not present, which is consistent with the decrease of the thermal properties of the composite as well as a more compact surface, as observed in the SEM. Therefore, it can be said that once the interaction between the ZnO-NPs and the C=O groups of the PCL has been completed, the degradation processes of the polymeric chain are stabilized, allowing a better interaction with the ZnO-NPs.
Figure 1. FT-IR spectrum for la formulation F3.
- DSC curves of the PCL/PLA/ZnO-NPs/TTEO nanocomposites. Why did the Tg point at 52 disappear for F2?
R// We appreciate the suggestion. Regarding the F2 endothermal peak, the smaller area might be due to the interaction between the carbonyl groups of the polymers and the ZnO-NPs, generating intermolecular spaces, which create gaps and rupture of the polymeric chain, causing the melting and crystallization processes to be affected. It can be observed that with an increase in the concentration of the nanoparticles, the exothermic peak of crystallization attributed to PLA decreases in the area, which means that not only affects the PCL as observed in F2, but an increase in concentration causes intramolecular interactions between the carbonyl groups of PLA and the nanoparticles.
- Figure 5. Morphology of PCL/PLA/ZnO-NPs blocks by SEM: More images with higher magnification should be added to observe the NPs.
R// Thank you very much for your comment. The magnification at which we work does not allow us to see the NPs. A TEM analysis might be helpful to observe the NPs in the scaffolds. However, grounding and cutting of the samples is needed. Still, we considered it unnecessary to demonstrate their presence using TEM, as the scaffold's properties reflected a well-dispersion. Due to a well-dispersion effect, it is impossible at the magnifications we are working with to observe NPs on the surface.
- Figure 8. Subdermal implantations of F2 at 60 days: Images are not colored.
R// Thank you very much for the observation. Figure 8A was changed.

Reviewer 3 Report
The manuscript reported the synthesis of four block formulations using polycaprolactone (PCL), polylactic acid (PLA), and zinc oxide nanoparticles (ZnO-NPs) for subdermal tissue regeneration. And their biocompatibility in vivo was studied. The morphology, thermal properties and in vivo degradation mechanism of PCL/PLA/ZnO-NPs blocks were reported. The results in vivo demonstrated the relevance and potential impact of the scaffolds in soft tissue regeneration. The manuscript is interesting. But the authors should pay attention to the following problems:
1.Why is the absorption peak of the stretching vibration of F3 at 3325 cm-1 for -OH so small in Figure1?
2.How to prove that ZnO NPs are loaded onto PCL/PLA? How are they distributed? For PCL/PLA/ZnO-NPs blocks, it is more informative to have EDX spectra of the different elements to determine whether they are homogeneous or not, there is no real information about their distribution.
3.What is the diffraction angle of the ZnO NPs in Fig. 2?
4.How do ZnO NPs affect the stabilization of PCL/PLA/ZnO-NPs blocks? This should be further explained in the manuscript.
5. How stable are the ZnO NPs loaded in PCL/PLA/ZnO-NPs blocks in the matrix? What is the ZnO NPs mechanism of action?
6. How are PCL/PLA/ZnO-NPs and ZnO NPs metabolized in the body? This should be explained better in the manuscript.
7. Several studies have been carried out on systems containing the three-component mixture PCL/PLA/ZnO NP, what are the innovations of the manuscript compared to this literature?
Please carefully check the manuscript for writing and grammar.
Author Response
We appreciate the comments from the reviewer. All the suggestions were addressed point by point for clarity.
Reviewer 3
The manuscript reported the synthesis of four block formulations using polycaprolactone (PCL), polylactic acid (PLA), and zinc oxide nanoparticles (ZnO-NPs) for subdermal tissue regeneration. And their biocompatibility in vivo was studied. The morphology, thermal properties, and in vivo degradation mechanism of PCL/PLA/ZnO-NPs blocks were reported. The results in vivo demonstrated the relevance and potential impact of the scaffolds in soft tissue regeneration. The manuscript is interesting. But the authors should pay attention to the following problems:
1.Why is the absorption peak of the stretching vibration of F3 at 3325 cm-1 for -OH so small in Figure1?
R// We appreciate your suggestion. Concerning formulation F3, the -OH band present in the composite (figure 1) decreases in intensity, probably due to the interaction between the ZnO-NPs and the carbonyl groups of the PCL, which promotes the destabilization of the polymeric matrix, generating acetaldehyde and the accumulation of lactide molecules from the oligomeric rings. Interestingly, once the concentration of ZnO-NPs is increased, the C=O vibrational mode attributed to PCL is not present, which is consistent with the decrease of the thermal properties of the composite as well as a more compact surface, as observed in the SEM. Therefore, it can be said that once the interaction between the ZnO-NPs and the C=O groups of the PCL has been completed, the degradation processes of the polymeric chain are stabilized, allowing a better interaction with the ZnO-NPs.
Figure 1. FT-IR spectrum for la formulation F3.
2.How to prove that ZnO NPs are loaded onto PCL/PLA? How are they distributed? For PCL/PLA/ZnO-NPs blocks, it is more informative to have EDX spectra of the different elements to determine whether they are homogeneous or not, there is no real information about their distribution.
R// Thank you very much for your comment. Our evidence for the presence of ZnO-NPs on the polymeric matrix decreases the thermal and crystalline properties obtained through thermogravimetric and XRD analysis. Now the distribution of the nanoparticles requires more robust equipment such as transmission electron microscopy, which will allow not only to show the presence of the nanoparticles but also their distribution in the scaffold.
3.What is the diffraction angle of the ZnO NPs in Fig. 2?
R// Thank you very much for your comment. The diffractogram in Figure 2 does not show the crystallographic peaks of the ZnO-NPs. However, it can be seen that as the concentration of the nanoparticles increases, the crystallinity of the composite decreases. This assumption is consistent with what has been observed in the other analyses since the nanoparticles interact with the polymers' carbonyl groups, promoting the generation of spaces between them.
4.How do ZnO NPs affect the stabilization of PCL/PLA/ZnO-NPs blocks? This should be further explained in the manuscript.
We appreciate your suggestion. According to the results, it has been concluded that ZnO-NPs interact with the C=O groups of the polymers present. It is observed that these nanoparticles interact primarily with the C=O groups of the PCL and then with those of the PLA, causing the formation of other compounds resulting from the degradation of the polymer. The information was added in several corrections as follows:
In the FTIR: lines 289-294
On the other hand, two vibrational modes at 1756 and 1724 cm-1 attributed to the carbonyl groups in F2 and F3 were observed, related to the PLA and PCL polymers [15,16]. For F4, only the peak at 1756 cm-1 was observed because the C=O groups interact with the ZnO-NPs promoting the accumulation of lactide molecules from the oligomeric ring and the acetaldehyde groups from the chain's degradation induced by ZnO-NPs [17].
In the XRD: lines 320-323
In this sense, it is observed that the intensity is preserved in the F1, F2, and F3 formulations, suggesting that the loss of crystallinity is more intense in F4 due to the generation of intermolecular spaces by the reduction of intramolecular interactions along the polymer chain [19].
- How stable are the ZnO NPs loaded in PCL/PLA/ZnO-NPs blocks in the matrix? What is the ZnO NPs mechanism of action?
R// Thank you very much for your suggestion. According to the literature, no mechanisms have been described in how nanoparticles interact with different polymeric matrices. It has been expressed through different analytical assumptions where ZnO-NPs are found within a polymeric matrix, causing the degradation of the polymeric matrix. However, our thermogravimetric tests do not show a degradation peak above 700°C because an oxidative medium is needed to see their degradation.
- How are PCL/PLA/ZnO-NPs and ZnO NPs metabolized in the body? This should be explained better in the manuscript.
R//We thank you for your appreciation. Both PCL and PLA have a similar mechanism of metabolization; in the first stage, the ester bonds are attacked by water molecules occurring in a hydrolytic degradation that fragments the polymeric chain allowing the phagocytosis of the smaller fragments. Still, this process is accompanied by enzymatic digestion originated by inflammatory cells; the ZnO nanoparticles to be incorporated into the material will be attacked and phagocytosed by inflammatory cells. In the manuscript, we can find it as follows: lines 580-586
The presence of the capsule may be related to the increased rate of resorption/degradation, as it has been reported that most biomaterials stimulate the fibrous capsule formation during tissue healing [20]. In the early stages of the healing process, a temporary matrix is formed from plasma proteins; later, the provisional matrix is replaced by a connective tissue matrix composed of type III and type I collagen fibers, and as the healing process progresses and the grafted material is reabsorbed, this matrix is reabsorbed and replaced by a connective tissue similar to the original tissue at the implantation site [21].
On the other hand, from line 457, the process for PLA was explained, and in the following paragraph, a line was added explaining that the mechanism for PCL was similar to that of PLA; concerning the ZnO-Nps, these dissociate on contact with water and can participate as a catalyst in the hydrolytic degradation of the polymeric matrix. Additionally, the cells are also able to eliminate by endocytosis some of these nanoparticles.
- Several studies have been carried out on systems containing the three-component mixture PCL/PLA/ZnO NP, what are the innovations of the manuscript compared to this literature?
R// We appreciate your observation. Although several studies contain this composite type and are applied in tissue engineering, these have only been evaluated under in vitro conditions using study cells. The innovation of this study lies in the subdermal implantation of a block-like morphology, i.e., in vivo studies using biomodels of Wistar rats, allowing an approach to medical devices according to the type of lesion in the human body.

Round 2
Reviewer 1 Report
The authors have responded to my comments and have addressed all my concerns, substantially improving the manuscript, therefore, I suggest publishing the paper in the current form.
Reviewer 2 Report
no comment